# Polarization insensitive frequency conversion for an atom-photon entanglement distribution via a telecom network

Rikizo Ikuta [1], Toshiki Kobayashi[1], Tetsuo Kawakami[1], Shigehito Miki[2], Masahiro Yabuno[2], Taro Yamashita[2,3], Hirotaka Terai[2], Masato Koashi[4], Tetsuya Mukai[5], Takashi Yamamoto[1] & Nobuyuki Imoto[1]

Long-lifetime quantum storages accessible to the telecom photonic infrastructure are essential to long-distance quantum communication. Atomic quantum storages have achieved subsecond storage time corresponding to 1000 km transmission time for a telecom photon through a quantum repeater algorithm. However, the telecom photon cannot be directly interfaced to typical atomic storages. Solid-state quantum frequency conversions fill this wavelength gap. Here we report on the experimental demonstration of a polarization-insensitive solid-state quantum frequency conversion to a telecom photon from a short-wavelength photon entangled with an atomic ensemble. Atom–photon entanglement has been generated with a Rb atomic ensemble and the photon has been translated to telecom range while retaining the entanglement by our nonlinear-crystal-based frequency converter in a Sagnac interferometer.

[1] Graduate School of Engineering Science, Osaka University, Toyonaka, Osaka 560-8531, Japan. [2] Advanced ICT Research Institute, National Institute of Information and Communications Technology (NICT), Kobe 651-2492, Japan. [3] Japan Science and Technology Agency, PRESTO, Kawaguchi, Saitama 332-0012, Japan. [4] Photon Science Center, Graduate School of Engineering, The University of Tokyo, Bunkyo-ku, Tokyo 113-8656, Japan. [5] NTT Basic Research Laboratories, NTT Corporation, Atsugi, Kanagawa 243-0198, Japan. Correspondence and requests for materials should be addressed to R.I. (email: ikuta@mp.es.osaka-u.ac.jp) or to N.I. (email: imoto@mp.es.osaka-u.ac.jp)

Quantum frequency conversion[1] (QFC) based on non-linear optical processes enables us to change the color of photons without destroying its quantum properties. This allows us to transfer quantum properties of a physical system to another one that has different accessible frequencies through a single photon[2–13]. Besides that, one could use QFC for other purposes such as erasing distinguishability of photons[14], manipulating spectral and temporal modes of photons[15–23], and performing frequency-domain quantum information processing[24–26] by tailoring of the pump light. Most of those abilities have been demonstrated with solid-state QFC devices because of its applicability to a wide frequency range, analogously to mirrors and beamsplitters (BSs) for the spatial manipulation of the photons.

The extension of the solid-state QFC for quantum storages has also been actively studied[27–29]. For long-distance quantum communication with quantum repeater algorithms[30–32], a long lifetime quantum storage system that can be entangled with a telecom photon is necessary. The cold Rb atomic ensemble is one of the promising quantum storage systems that has a long lifetime and a high efficiency atom–photon entanglement generation[3,33–38]. Recently, solid-state QFC of a single photon from a cold Rb atomic ensemble has been demonstrated[28,29], in which a non-classical correlation between atoms and a telecom photon was reported[28] and a single spin excitation of atoms heralded by a telecom photon detection was observed[29]. But the quantum state preservation, which is an ability that cannot be mimicked by a classical memory, has never been shown yet.

In this paper, we report a polarization-insensitive QFC (PIQFC), which converts the frequency (wavelength) of a photon while preserving the input polarization state. Our solid-state PIQFC device consists of a waveguided periodically poled lithium niobate (PPLN) crystal installed in a Sagnac interferometer. By using the QFC device, we converted a 780-nm polarized photon entangled with a cold Rb atomic ensemble to a telecom wavelength of 1522 nm. Entanglement between the Rb atoms and the converted telecom photon has been clearly observed. The demonstration of interface between telecom photons and a good quantum storage is a key ingredient in the future quantum network technology.

## Results

**Polarization-insensitive QFC.** We first review the conventional QFC of a single-mode light with a specific polarization based on the second-order nonlinear optical effect[1,6]. When a pump light at angular frequency $\omega_p$ is sufficiently strong, the Hamiltonian of the process is described by $H = i\hbar \xi^* a_l^\dagger a_u + \text{h.c.}$, where h.c. represents the Hermitian conjugate of the first term, and $a_u$ and $a_l$ are annihilation operators of upper and lower frequency modes at angular frequencies $\omega_u$ and $\omega_l (= \omega_u - \omega_p)$, respectively. Coupling constant $\xi = |\xi| e^{i\phi}$ is proportional to the complex amplitude of the pump light with its phase $\phi$.

As shown in Fig. 1d, when two QFCs working for two different polarization modes are superposed, the interaction Hamiltonian can be described by $H = i\hbar \left( \xi_H^* a_{l,H}^\dagger a_{u,H} + \xi_V^* a_{l,V}^\dagger a_{u,V} \right) + \text{h.c.}$, where $a_{u(l),H}$ and $a_{u(l),V}$ are annihilation operators of horizontally (H) polarized and vertically (V) polarized upper(lower) frequency modes, respectively. $\xi_{H(V)} = \left| \xi_{H(V)} \right| e^{i\phi_{H(V)}}$ is proportional to the amplitudes of the H(V)-polarized pump light with phase $\phi_{H(V)}$. By using the Heisenberg representation, annihilation operators $a_{u,H(V),\text{out}}$ and $a_{l,H(V),\text{out}}$ of the upper and lower frequency modes coming from the nonlinear optical medium are represented by

$$a_{u,H(V),\text{out}} = t_{H(V)} a_{u,H(V)} - r_{H(V)} a_{l,H(V)} \qquad (1)$$

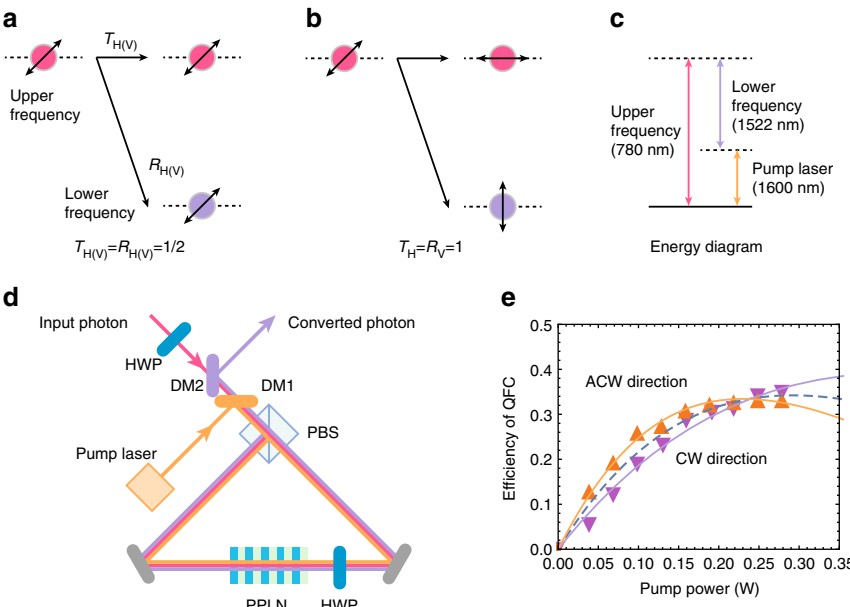

**Fig. 1** QFC as a frequency-domain linear optics with the two polarization modes. **a** Concept of a non-polarizing frequency-domain half BS for $T_{H(V)} = R_{H(V)} = 1/2$. **b** A frequency-domain PBS for $T_H = R_V = 1$ and $T_V = R_H = 0$. **c** Energy diagram related to the QFC used in our experiment. The difference frequency generation of a photon from 780 nm (upper frequency) to 1522 nm (lower frequency) is performed by using a strong pump light at 1600 nm. **d** The experimental setup of dual-polarization-mode QFC. Type-0 quasi-phase-matched PPLN crystal as a nonlinear optical medium which acts on only the V-polarized photons is installed in the Sagnac type interferometer. The detailed explanation of the QFC is in the main text. DM dichroic mirror, HWP half wave plate, PBS polarizing beamsplitter. **e** Conversion efficiency from the front to the end of the QFC. It is measured by using classical laser light and a power meter. The error of the power meter was about 1% and they are smaller than the markers of data. Solid curves are fitted to data for CW and ACW directions separately by using functions $\eta_{\max} \sin^2 \left( \sqrt{gP} \right)$, where $P$ is the pump power. Parameters $\eta_{\max}$ and $g$ for CW(ACW) directions are estimated as 0.39 (0.33) and 5.8/W (10.7/W), respectively. A broken curve is fitted to two data set together, resulting in $\eta_{\max} = 0.34$ and $g = 8.5$/W

and

$$a_{\text{l,H(V),out}} = r^*_{\text{H(V)}} a_{\text{u,H(V)}} + t_{\text{H(V)}} a_{\text{l,H(V)}}, \tag{2}$$

where $t_{\text{H(V)}} = \cos\left(\left|\xi_{\text{H(V)}}\right|\tau\right)$ and $r_{\text{H(V)}} = e^{i\phi_{\text{H(V)}}}\sin\left(\left|\xi_{\text{H(V)}}\right|\tau\right)$. $\tau$ is the traveling time of the light pulses through the medium. The transmittance $T_{\text{H(V)}} \equiv \left|t_{\text{H(V)}}\right|^2$ and the reflectance $R_{\text{H(V)}} \equiv \left|r_{\text{H(V)}}\right|^2$ for splitting into the two frequency modes propagating in the same direction can be changed by adjusting the amplitudes of the pump light. When $T_{\text{H}} = T_{\text{V}}$ and $R_{\text{H}} = R_{\text{V}}$ are satisfied, and a single photon converted event is postselected, the QFC process while preserving the input polarization state up to the constant phase shift of $\phi_{\text{H}} - \phi_{\text{V}}$ is achieved. By compensating the constant phase, we thus achieve PIQFC, which we used in the experiments reported in this paper. On the other hand, we may also achieve other modes of operation depending on the choice of $R_{\text{H}}$ and $R_{\text{V}}$. (I) For $T_{\text{H(V)}} = R_{\text{H(V)}} = 1/2$, it becomes a non-polarizing frequency-domain half BS[24–26,39] (see Fig. 1a). (II) For $T_{\text{H}} = R_{\text{V}} = 1$ and $T_{\text{V}} = R_{\text{H}} = 0$, it becomes a frequency-domain polarizing BS (PBS) (see Fig. 1b). (III) For $T_{\text{H}} \neq T_{\text{V}}$ and $0 < T_{\text{H(V)}} < 1$, it becomes a frequency-domain partially polarizing BS (PPBS). PPBSs with proper settings of the transmittance and the reflectance can be used to perform frequency-domain quantum information protocols, such as entanglement distillation[40,41], probabilistic nonlinear optical gate[42,43], quantum state estimation[44], and manipulation of multipartite entangled states[45,46].

We explain the experimental detail of the PIQFC in Fig. 1d. The nonlinear optical medium for QFC is a type-0 quasi-phase-matched PPLN waveguide (also see Methods section) that converts a V-polarized input photon to a V-polarized photon

with the use of the V-polarized pump light. The PPLN is installed in a Sagnac interferometer. In this demonstration, we prepare a polarized upper frequency photon at 780 nm entangled with a Rb atomic ensemble as an input signal to the converter that we explain in detail later. As shown in Fig. 1c, by using a strong pump light at 1600 nm with a linewidth of 150 kHz, the upper frequency photon at 780 nm is converted to the lower frequency photon at 1522 nm by difference frequency generation. In Fig. 1d, the polarization of the input photon is flipped from H (V) polarization to V (H) by a half wave plate (HWP). The photon is combined with the diagonally polarized strong pump light at 1600 nm at a dichroic mirror (DM1). At a PBS, the H- and V-polarized components of them are split into clockwise (CW) and anti-clockwise (ACW) directions, respectively. For the CW direction, after flipping from H to V polarization at a HWP, the V-polarized input photon and the pump light are coupled to the PPLN waveguide. After the conversion, the V-polarized photons and pump light are reflected by the PBS. Then, only the converted photon is extracted from the reflection port of DM2, being separated from the pump light and the residual input photon by DM1 and DM2, respectively. On the other hand, for the ACW direction, the V-polarized input photon and the pump light are coupled to the PPLN waveguide. After the conversion, the polarization is flipped from V to H by the HWP, and the photons and the pump light pass through the PBS. Finally, only the converted photon is extracted by DM1 and DM2. The conversion efficiencies of the QFC for CW and ACW directions are shown in Fig. 1e (see Methods section).

**Experimental setup**. In order to prepare a 780-nm signal photon entangled with the Rb atoms, we construct an experimental setup as shown in Fig. 2a. We use Λ-type energy levels of D$_2$ line at 780

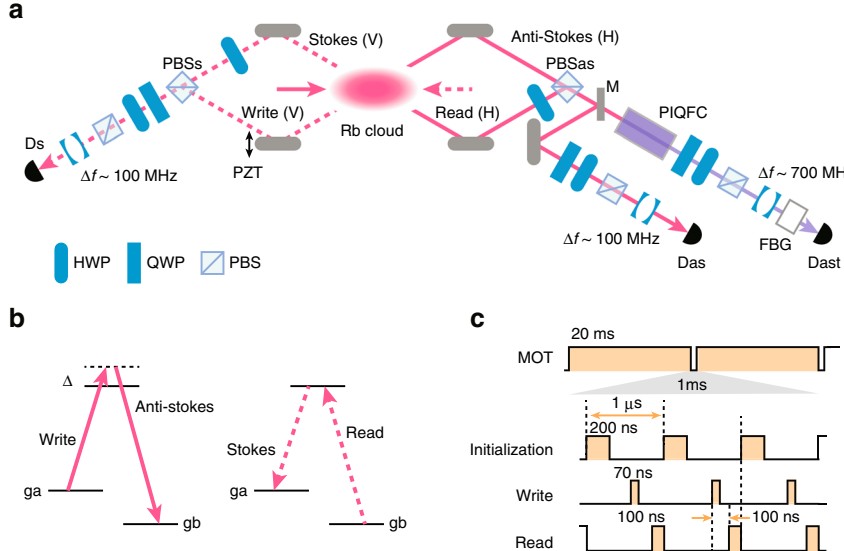

**Fig. 2** Experimental setup. **a** Our experimental setup for entanglement between Rb atoms and visible/telecom photons without/with QFC. When mirror M is flipped up, AS photon is detected by $D_{\text{as}}$ without QFC. When mirror M is flipped down, AS photon is input to polarization-insensitive QFC (PIQFC) as shown in Fig. 1d and then the converted photon is detected by $D_{\text{ast}}$. In order to stabilize the interferometer in optical paths of Stokes and anti-Stokes photons, we use a conventional diode laser at a center wavelength of 850 nm (not shown). The light enters into the interferometer from the vacuum port of PBS$_{\text{s}}$, and then after passing along the same four optical paths for Stokes and anti-Stokes photons, it comes from the vacuum port of PBS$_{\text{as}}$. The light is detected by a photo detector, and its detection signal is used for feedback control of the interferometer by a mirror on a Piezo stage (PZT). **b** Λ-type energy levels of D$_2$ line in $^{87}$Rb used in our experiment. The ground levels of $g_a$ and $g_b$ correspond to the levels of $5^2S_{1/2}$, $F = 2$ and $5^2S_{1/2}$, $F = 1$, respectively. The excited level is $5^2P_{3/2}$, $F' = 2$. $\Delta$ is set to ~10 MHz. The magnetic sublevel is degenerated in our experiment due to the absence of the magnetic field, and the polarization selection of write, read, AS and S light is to extract larger amounts of the emitted photons under an assumption that the initialized atomic states in $g_a$ are fully mixed over the magnetic sublevels (Supplemental Material in ref. [29]). **c** Time sequence of the experiment. The quantum experiment is performed within 1 ms during the MOT is turned off. The period of the two initialization pulses is 1 μs and the injection of them is repeated 990 times within 1 ms

nm in $^{87}$Rb atoms ($5^2S_{1/2} \leftrightarrow 5^2P_{3/2}$) as shown in Fig. 2b. We prepare the Rb atomic ensemble by a magneto-optical trap (MOT) in 20 ms (see Fig. 2c). After the trapping lasers and the magnetic field for the MOT are turned off, we perform the QFC experiment 990 times within 1 ms. A horizontally H-polarized 200-ns initialization pulse at the resonant frequency between a ground level $g_b$ ($F = 2$) and an excited level ($F' = 2$) initializes the atoms into another ground level $g_a$ ($F = 1$). Then a vertically V-polarized 70-ns write pulse blue-detuned by $\Delta \sim 10$ MHz from the resonant frequency between $g_a$ and the excited level is injected to the atoms, causing the Raman transition from $g_a$ to $g_b$ with emission of anti-Stokes (AS) photons.

The momentum conservation guarantees that the wave vector $\mathbf{k}_{atom}$ of the collective spin excitation of the atoms satisfies $\mathbf{k}_{atom} = \mathbf{k}_W - \mathbf{k}_{AS}$, where $\mathbf{k}_{W/AS}$ is the wave vector of the write/AS light. When we postselect a particular wave vector (path) of a single AS photon in a single mode whose quantum state is denoted by $\left|path_x\right\rangle_{AS}$, the wave vector of the atoms corresponding to the photonic state is decided by the momentum conservation. We denote the atomic state by $|k_x\rangle_{atom}$. As a result, when we postselect two different wave vectors of the AS photon corresponding to the states $\left|path_+\right\rangle_{AS}$ and $\left|path_-\right\rangle_{AS}$, we obtain a quantum state of the atoms and the AS photon as

$$\alpha\left|path_+\right\rangle_{AS}\left|k_+\right\rangle_{atom} + \beta\left|path_-\right\rangle_{AS}\left|k_-\right\rangle_{atom}, \quad (3)$$

with $|\alpha|^2 + |\beta|^2 = 1$. The subscripts $+$ and $-$ imply the upper and lower optical paths of the AS photons in Fig. 2a. By adjusting the excitation probabilities such that $\alpha = \beta$ is satisfied, we obtain a maximally entangled state. In our experiment, we select the H-polarized AS photons emitted in two directions at small angles ($\sim \pm 3°$) relative to the direction of the write pulse, whose emission probabilities are the same ideally. By using a HWP and a PBS (PBS$_{as}$), path information of AS photons is transformed to polarization information. After the operation, $\left|path_{+/-}\right\rangle_{AS}$ in Eq. (3) is changed to the H/V-polarized states denoted by $|H/V\rangle_{AS}$, and we obtain

$$\left(|H\rangle_{AS}|k_+\rangle_{atom} + |V\rangle_{AS}|k_-\rangle_{atom}\right)/\sqrt{2}. \quad (4)$$

In order to evaluate the quantum correlation between the atoms and the photons, we inject an H-polarized 100-ns read light at the resonant frequency between $g_b$ and the excited level into the atoms. The read light provides the transition of the Rb atoms to $g_a$ and generation of the Stokes (S) photons. In our experiment, we collect only the V-polarized component of the S photons. Because of the momentum conservation, wave vector $\mathbf{k}_S$ of the S photons satisfies $\mathbf{k}_S = \mathbf{k}_{atom} + \mathbf{k}_R$. The direction of the emitted S photons is decided by the wave vector of the atomic excitation. Because such a read operation does not access the AS photon, the operation never increase or newly create the entanglement between the atoms and the AS photon. Thus observation of an entangled state of the which-path state of the S photon and the polarizing AS photon after the read operation is the evidence of the entanglement between the atoms and the AS photon before the read operation.

In the experiment, we inject the read pulse from the direction opposite to the write pulse. The wave vector $k_R$ of the read pulse satisfies $\mathbf{k}_R \sim -\mathbf{k}_W$, leading to the relation $\mathbf{k}_S \sim -\mathbf{k}_{AS}$ from the momentum conservation. This means the S photons are emitted in a direction at $\sim \mp 3°$ relative to the direction of the read pulse when the AS photons are emitted in a direction at $\sim \pm 3°$ relative to that of the write pulse. By using a HWP and a PBS (PBS$_s$) shown

in Fig. 2a, the path information of the V-polarized S photons is transformed into the polarization. Finally, we can observe the entanglement between the atoms and the AS photons through the polarization entangled photon pair of the AS and the S photons.

After passing through a polarization analyzer composed of a QWP, a HWP, and a PBS for the quantum state tomography[47], S photon passes through a monolithic cavity-coated lens[48] as a frequency filter with an observed bandwidth of 78 MHz and is coupled to a single-mode optical fiber. Then S photon is detected by a silicon avalanche photon detector (APD) denoted by $D_s$ with a quantum efficiency of ~60%.

When we do not perform QFC, AS photon is detected by another APD ($D_{as}$) after passing through a polarization analyzer, a cavity-coated lens with an observed bandwidth of 61 MHz and a single-mode optical fiber. When we perform QFC, mirror M in Fig. 2a is flipped down in order to send the AS photon to the PIQFC depicted in Fig. 1d. We set the conversion efficiency from the front of the PPLN waveguide to the end of the QFC to 33% by using the pump power of 0.23 and 0.21 W for CW and ACW directions, respectively (see Fig. 1e). The telecom photon from the QFC passes through a polarization analyzer followed by an etalon with a bandwidth of 690 MHz and a pair of fiber Bragg gratings with a total bandwidth of ~0.9 GHz. Finally, the telecom photon is detected by a superconducting single photon detector (SSPD) denoted by $D_{ast}$ with a quantum efficiency of ~60%[49].

We repeat the above measurement about 47,000 times per second. We use a trigger signal for starting each sequence as a start signal of a time-to-digital converter (TDC). The photon countings measured by $D_s$, $D_{as}$ and $D_{ast}$ are used as stop signals of the TDC. We collect the coincidence events between the signals of modes S and AS in their time windows of 64 ns.

**Experimental results**. Without QFC, we performed the quantum state tomography between the S photon and the AS photon and reconstructed density operator $\rho_{S,AS}$ by the use of the iterative maximum likelihood method[50]. We estimated entanglement of formation (EoF)[51] $E$ and the purity defined by $P = \mathrm{tr}\left(\rho_{S,AS}^2\right)$ as $E = 0.37 \pm 0.11$ and $P = 0.61 \pm 0.06$, respectively. We also estimated a maximized fidelity to a maximally entangled state $U_\theta|\phi^+\rangle$ defined by $F = \max_\theta\langle\phi^+|U_\theta^\dagger\rho_{S,AS}U_\theta|\phi^+\rangle$, whose value was $F = 0.78 \pm 0.05$ for $\theta = \theta_0 = -65°$. Here $|\phi^+\rangle = \left(|H\rangle_{AS}|H\rangle_S + |V\rangle_{AS}|V\rangle_S\right)/\sqrt{2}$ and $U_\theta = \exp(-i\theta Z/2) \otimes I$ with $Z = |H\rangle\langle H| - |V\rangle\langle V|$ and $I = |H\rangle\langle H| + |V\rangle\langle V|$. The calculated values of the matrix elements of density operator $U_{\theta_0}^\dagger\rho_{S,AS}U_{\theta_0}$ and its visualization are shown in Table 1 and Fig. 3, respectively. These results show the entanglement between AS photon and the Rb atoms. The observed count rate of the two-photon state was about 0.08 Hz through the overall experiment time of 16 h including the load time of the atoms by MOT.

With QFC, we performed the quantum state tomography between S photon and the wavelength-converted AS photon. The estimated EoF and purity of reconstructed density operator $\rho_{S,ASt}$ were $E = 0.25 \pm 0.13$ and $P = 0.55 \pm 0.07$, respectively. The maximized fidelity to $U_\theta|\phi^+\rangle$ about $\theta$ was $F = 0.69 \pm 0.07$ for $\theta = \theta_1 = 93°$. The calculated values of the matrix elements of density operator $U_{\theta_1}^\dagger\rho_{S,ASt}U_{\theta_1}$ and its visualization are shown in Table 2 and Fig. 4, respectively. The EoF $E$ is clearly $>0$, which shows that the state of the Rb atoms and the telecom photon has entanglement. From the result, we succeeded the creation of the entanglement between the Rb atoms and the telecom photon by using the PIQFC. The observed count rate of the two-photon state was about 0.0065 Hz through the experiment time of 83 h.

**Table 1 The elements of the density matrix without QFC**

|  | $|HH\rangle$ | $|HV\rangle$ | $|VH\rangle$ | $|VV\rangle$ |
|---|---|---|---|---|
| $\langle HH|$ | 0.42 | $-0.05 + 0.03i$ | $-0.03 + 0.07i$ | 0.34 |
| $\langle HV|$ | $-0.05$ to $0.03i$ | 0.04 | $-0.04i$ | $0.01 + 0.01i$ |
| $\langle VH|$ | $-0.03$ to $0.07i$ | $0.04i$ | 0.08 | $-0.07 + 0.03i$ |
| $\langle VV|$ | 0.34 | $0.01$ to $0.01i$ | $-0.07-0.03i$ | 0.46 |

The values are calculated from $\left\langle mn\left|U_{\theta_0}^\dagger\rho_{S,AS}U_{\theta_0}\right|m'n'\right\rangle$ for $m, n, m', n' = \{H, V\}$

**Table 2 The elements of the density matrix with QFC**

|  | $|HH\rangle$ | $|HV\rangle$ | $|VH\rangle$ | $|VV\rangle$ |
|---|---|---|---|---|
| $\langle HH|$ | 0.43 | $-0.07$ | $-0.02$ | 0.25 |
| $\langle HV|$ | $-0.07$ | 0.07 | $0.04i$ | $-0.03$ to $0.02i$ |
| $\langle VH|$ | $-0.02$ | $-0.04i$ | 0.04 | $-0.01$ to $0.03i$ |
| $\langle VV|$ | 0.25 | $-0.03 + 0.02i$ | $-0.01 + 0.03i$ | 0.46 |

The values are calculated from $\left\langle mn\left|U_{\theta_1}^\dagger\rho_{S,ASt}U_{\theta_1}\right|m'n'\right\rangle$ for $m, n, m', n' = \{H, V\}$

## Discussion

First, let us discuss the reasons for the degradation of the fidelity of the reconstructed state due to QFC by comparing the cross correlation functions without QFC and those with QFC. We define the normalized cross correlation function without(with) QFC by $g_{S,AS|mn}^{(2)}\left(g_{S,ASt|mn}^{(2)}\right)$ which is measured at $D_{as(ast)}$ and $D_s$ with $m$ and $n$ polarization, respectively, where $m, n \in \{H, V\}$. The cross correlations are calculated from the main experimental data of the coincidence counts used in the reconstruction of $\rho_{S,AS}$ and $\rho_{S,ASt}$, and the single counts that were recorded in the same runs. We list the estimated values in Table 3. In order to discuss genuine effect of QFC, we should be careful about the effect of the strong laser light at 850 nm used for stabilizing the interferometer. The main cause of the degradation of the fidelity without QFC is the non-negligible contamination from the 850-nm light used for the stabilization of the interferometer by the mirror on PZT as mentioned in the caption of Fig. 2a. On the other hand, the fidelity with QFC does not suffer: the 850-nm light is eliminated because it does not satisfy the phase matching of the PPLN crystal or the converted light cannot pass through the frequency filters even if the light is converted. For this reason, we ran additional experiments for measuring the intensity correlation $g_{S,AS|mn}^{(2)*}$ without QFC while turning off the 850-nm laser light, whose result is shown in Table 3. For $(m, n) = (H, V)$ and $(V, H)$, the cross correlation functions are around unity, which means the S photons (Rb atoms) and AS photons have no correlation. By using the values for $(m, n) = (H, H)$ and $(V, V)$, we will estimate the amount of the 850-nm noise photons in the experiment without QFC, and will thereby estimate the amount of noise photons produced in the process of QFC. It is natural to assume the S photons and the noise photons are statistically independent. In this case the following equations are satisfied[29]:

$$g_{S,ASt|mn}^{(2)} = \frac{g_{S,AS|mn}^{(2)*}\zeta_n + 1}{\zeta_n + 1}, \quad (5)$$

$$g_{S,AS|mn}^{(2)} = \frac{g_{S,AS|mn}^{(2)*}\zeta_n' + 1}{\zeta_n' + 1}, \quad (6)$$

where $\zeta_n$ and $\zeta_n'$ are the signal-to-noise ratios, $\zeta_n$ is ratio of the average photon number of the AS photons just before the QFC at

mirror M and the equivalent input noise to the QFC device, and $\zeta_n'$ is ratio of that of the AS photons and the 850-nm noise photons at mirror M. From the equations, we obtain $\zeta_H = 0.56$ and $\zeta_V = 0.65$ with QFC and $\zeta_H' = 0.46$ and $\zeta_V' = 0.44$ without QFC.

The above experimental results imply that the signal-to-noise ratios are almost polarization independent. In the following discussion, we assume the signal-to-noise ratios are $\zeta_{QFC} = 0.6$ with QFC and $\zeta_{850} = 0.45$ without QFC for any polarization. Here, we estimate an expected polarization correlation without QFC which would be obtained if we could remove the contamination from the 850-nm laser without affecting the phase stability. Let us consider the three visibilities, $V_z(\rho) = \text{tr}(\rho ZZ)$, $V_x(\rho) = \text{tr}(\rho XX)$ and $V_y(\rho) = \text{tr}(\rho YY)$, where $\rho$ is a density operator, $X = |H\rangle\langle V| + |V\rangle\langle H|$ and $Y = -i|H\rangle\langle V| + i|V\rangle\langle H|$. The fidelity of state $\rho$ to the maximally entangled state $|\phi^+\rangle\langle\phi^+| = (1 + XX)(1 + ZZ)/4$ is then given by $F(\rho) = (1 + V_z(\rho) + V_x(\rho) - V_y(\rho))/4$. For the reconstructed density operator $U_{\theta_0}^\dagger\rho_{S,AS}U_{\theta_0}$ without QFC, the visibilities are $V_z = 0.76$, $V_x = 0.68$, and $V_y = -0.69$, respectively. By using these visibilities, $g_{S,AS|mn}^{(2)*}$ and $\zeta_{850}$, the expected visibilities without QFC when we could remove the contamination of 850-nm light are estimated as $V_z^* = 0.90$, $V_x^* = 0.81$, and $V_y^* = -0.82$. The corresponding fidelity is $F^* = \left(1 + V_z^* + V_x^* - V_y^*\right)/4 = 0.88$. The details of the calculation are shown in Methods section.

Similarly to the above calculation, we can calculate expected degradation of the visibilities from $V_i^*$ for $i = X, Y, Z$ associated with the noise photons produced in QFC, using the value of $\zeta_{QFC}$. The estimated visibilities are $V_z^t = 0.76$, $V_x^t = 0.68$, and $V_y^t = -0.69$. On the other hand, the visibilities calculated from the density operator $U_{\theta_1}^\dagger\rho_{S,ASt}U_{\theta_1}$ reconstructed in the actual experiment with QFC are $V_z^{t,obs} = 0.79$, $V_x^{t,obs} = 0.50$, and $V_y^{t,obs} = -0.49$. We see that $V_z^{t,obs}$ is close to $V_z^t$, but $V_y^{t,obs}$ is significantly lower than $V_{x(y)}^t$. This fact implies that we may ascribe the degradation of fidelity with QFC to the two causes, the noise photons induced from the strong pump for QFC and decoherence between the H- and V-polarized photons.

Based on the above analysis, we discuss possible improvements of the fidelity after QFC. As for the noise photons, the small value of $\zeta_{QFC}$ comes from the fact that the collection efficiency for the AS photons is <1% in the current experiment. If we could improve $\zeta_{QFC}$ 10 times enhancing it to around ~6 by increasing the collection efficiency of the AS photons (which could be done with state-of-the-art technologies[52]), we would expect $V_z^{t,exp} = 0.89 (\sim V_z^*)$, $V_x^{t,exp} = 0.56$, and $V_y^{t,exp} = -0.55$, resulting in an expected improvement of the fidelity to $F = 0.75$. In order to improve it further to approach $F^*$, it is necessary to mitigate the decoherence. We speculate that the cause of the decoherence is the phase fluctuations in the interferometers of the S and AS photons due to the fluctuation or drift of the wavelength of the 850-nm laser, since the duration of the experimental run with QFC is much longer than the run without QFC. If this turned out

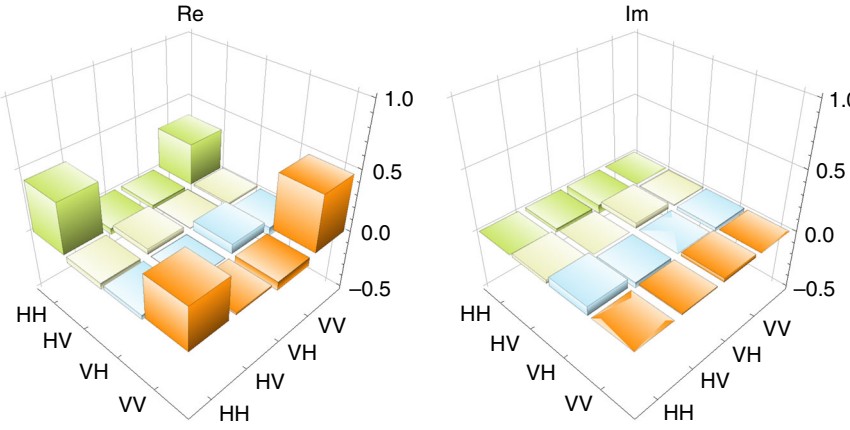

**Fig. 3** Visualization of the density matrix without QFC. 'Re' and 'Im' are the real and the imaginary parts of $U_{\theta_0}^{\dagger}\rho_{S,AS}U_{\theta_0}$, respectively

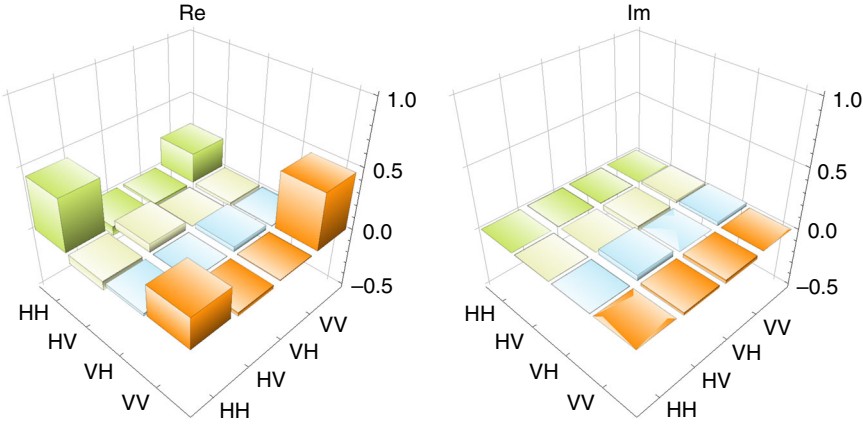

**Fig. 4** Visualization of the density matrix with QFC. 'Re' and 'Im' are the real and the imaginary parts of $U_{\theta_1}^{\dagger}\rho_{S,ASt}U_{\theta_1}$, respectively

**Table 3 The cross correlation functions.** $g_{S,AS|mn}^{(2)}$ **and** $g_{S,ASt|mn}^{(2)}$ **are calculated from the main experimental data without and with QFC, respectively, while turning on the 850-nm laser light**

| $(m, n)$ | (H, H) | (H, V) | (V, H) | (V, V) |
|---|---|---|---|---|
| $g_{S,AS|mn}^{(2)*}$ | 29.0 ± 4.1 | 0.5 ± 0.5 | 1.4 ± 0.8 | 15.7 ± 2.4 |
| $g_{S,AS|mn}^{(2)}$ | 9.8 ± 0.9 | 0.9 ± 0.3 | 1.0 ± 0.5 | 5.5 ± 0.5 |
| $g_{S,ASt|mn}^{(2)}$ | 11.0 ± 1.4 | 1.4 ± 0.5 | 0.7 ± 0.3 | 6.8 ± 0.9 |

$g_{S,AS|mn}^{(2)*}$ is measured without QFC while turning off the 850-nm laser light. The attached errors are the standard deviations calculated with an assumption of the Poissonian statistics of the photon counts

to be the case, a possible solution could simply be a more stable laser.

In conclusion, we have shown entanglement between the wave vector of collective spin excitation of Rb atoms and a polarized telecom photon by using PIQFC composed by a PPLN waveguide installed in a Sagnac interferometer. Recent researches showed efficient and sub-second lifetime quantum memory[38] and a multiplexed quantum memory[53,54] by using Rb atomic ensembles. Combining such state-of-the-art technologies with our experimental result will be useful for fiber-based quantum communication over long distance. Furthermore, the PIQFC is applicable to various conversion systems for matter-based

quantum storages, such as trapped ions[55,56], diamond color centers[39], and quantum dots[10], which are useful for measurement-based quantum computation[57–59]. The potential of our QFC is not limited by the use of the PIQFC. The proposed dual-polarization-mode QFC has additional features as a non-polarizing frequency-domain half BS, a frequency-domain PBS, and a frequency-domain PPBS. The devices will provide various kinds of tasks developed in the linear optical quantum information processing.

During preparation of our paper, we found a related work[60] in which polarization entanglement between a trapped calcium ion and a telecom O-band photon with QFC was demonstrated.

## Methods
**PPLN waveguide**. The PPLN waveguide used in our experiment is a ridged type with 8-μm wide. The length is 40 mm. The temperature is controlled to be about 20 °C for the best conversion efficiency.

The dependencies of the conversion efficiencies on the pump power for CW and ACW directions shown in Fig. 1e are measured by using laser light, which propagates along the same spatial path as the AS photon. The conversion efficiency was calculated by measuring the power of the 780-nm light in front of the QFC and the 1522-nm light at end of the QFC before the fiber coupling. The maximum conversion efficiency is not unity. One of the reasons is the non-unit coupling efficiencies of the signal light, which are 0.83 and 0.65 for CW and ACW directions, respectively. Another reason may be mode mismatching of the signal and the pump light propagating in the PPLN waveguide. We aligned the QFC circuit to achieve the same maximum conversion efficiency in total for CW and ACW directions.

We estimated the amount of the noises at QFC by measuring the photon counts without the input of the signal photons. The count rates were $3.1 \times 10^{-6}$ counts/pulse and $3.5 \times 10^{-6}$ counts/pulse for H and V polarization, respectively. By considering the quantum efficiency 60% of the SSPD and transmittances 50 and

17% of the frequency filters (the etalon and the fiber Bragg gratings (FBGs)), the noise photons just after the QFC are estimated to be $6.1 \times 10^{-5}$ and $7.0 \times 10^{-5}$ counts/pulse for H and V polarization, respectively, in the bandwidth of 540 MHz that are calculated by using the bandwidths 690 MHz and 0.9 GHz of the frequency filters. Since the total conversion efficiency used in our experiment is 0.33 for both H and V polarization, the signal-to-noise ratios for a deterministic single photon input are 5400 and 4700, respectively.

**Transmittance of the optical components**. Below we list the transmittance of the optical components for the signal photons in our experiment. For 780 nm, the transmittances of DM1 and DM2 in the QFC circuit are both 0.99. The transmittances of the monolithic cavity-coated lenses for the S and AS photons are both 0.2 including the fiber coupling efficiency.

For 1522 nm, the transmittance of DM1 is over 0.98 and the reflectances of DM2 are about 1.0 and 0.88 for H- and V-polarized photons, respectively, which are included in the efficiency of the QFC shown in Fig. 1e. The transmittance of the etalon is 0.5 inclusive of the fiber coupling efficiency. The transmittance of a pair of FBGs is 0.17.

**Estimation of visibilities and fidelity**. As is the common situation in quantum information experiments using single photons, we assume that $\langle mn|\rho|mn\rangle$ is measured by coincidence probability $P_{mn}$ of the $m$ and $n$ polarized two photons[47]. In this case, the visibility $V_z$ is written as $V_z = (P_{HH} + P_{VV} - P_{HV} - P_{VH})/(P_{HH} + P_{VV} + P_{HV} + P_{VH})$. When the single counts are independent of the measurement polarization, the visibilities are calculated by using the cross correlation functions for $m$ and $n$ polarization. For example, $V_z^*$ is given by

$$V_z^* = \frac{g_{S,AS|HH}^{(2)*} + g_{S,AS|VV}^{(2)*} - g_{S,AS|HV}^{(2)*} - g_{S,AS|VH}^{(2)*}}{\sum\limits_{m,n\in\{H,V\}} g_{S,AS|mn}^{(2)*}}. \tag{7}$$

From Eq. (6), we have

$$V_z = \frac{g_{S,AS|HH}^{(2)} + g_{S,AS|VV}^{(2)} - g_{S,AS|HV}^{(2)} - g_{S,AS|VH}^{(2)}}{g_{S,AS|HH}^{(2)} + g_{S,AS|VV}^{(2)} + g_{S,AS|HV}^{(2)} + g_{S,AS|VH}^{(2)}} \tag{8}$$
$$= V_z^* \gamma^*(\zeta_{850})$$

with

$$\gamma^*(\zeta) := \left(1 + \frac{4}{\zeta \sum\limits_{m,n\in\{H,V\}} g_{S,AS|mn}^{(2)*}}\right)^{-1}. \tag{9}$$

From $g_{S,AS|HH}^{(2)*} = 29.0$, $g_{S,AS|VV}^{(2)*} = 15.7$ and $g_{S,AS|HV}^{(2)*} = g_{S,AS|VH}^{(2)*} = 1$, we obtain $\sum_{m,n\in\{H,V\}} g_{S,AS|mn}^{(2)*} = 46.7$, and this is basis independent due to the property of the trace. Together with our assumption that $\zeta_{850}$ is polarization independent, we also have $V_{x(y)} = V_{x(y)}^* \gamma^*(\zeta_{850})$. Using the values of $V_z = 0.76$, $V_x = 0.68$, and $V_y = -0.69$, we obtain $V_z^* = 0.90$, $V_x^* = 0.81$, and $V_y^* = -0.82$. In a similar manner, we also obtain $V_z^t = V_z^* \gamma^*(\zeta_{QFC}) = 0.76$, $V_x^t = V_x^* \gamma^*(\zeta_{QFC}) = 0.68$, and $V_y^t = V_y^* \gamma^*(\zeta_{QFC}) = -0.69$.

**Data availability**. The data that support the findings of this study are available from the corresponding author on reasonable request.

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

## Acknowledgements

We thank Yoshiaki Tsujimoto and Motoki Asano for helpful discussions about QFC. This work was supported by CREST, JST JPMJCR1671 and MEXT/JSPS KAKENHI Grant Numbers JP26286068, JP15H03704, JP16H02214, and JP16K17772.

## Author contributions

R.I., T.K., T.Y., and T.M. designed the experiment. R.I., T.K., and T.K. carried out the experiments under supervision of T.Y., M.K., T.M., and N.I. S.M., M.Y., T.Y., and H.T. developed the system of the superconducting single-photon detectors. All authors analyzed the experimental results, contributed to the discussions and interpretations. R.I. wrote the manuscript, with inputs from all coauthors.

## Additional information

**Competing interests:** The authors declare no competing interests.

