## [Peer Review File · Nature Communications]

Reviewers' comments:

Reviewer #1 (Remarks to the Author):

In their manuscript, entitled 'Polarization insensitive frequency conversion for an atom-photon entanglement distribution via a telecom network', the authors show entanglement between a cold atomic ensemble and a frequency converted photon at telecom wavelength. The authors first introduce the concept of their polarization-independent quantum frequency converter (PIQFC) based on a PPLN waveguide placed inside a Sagnac interferometer. They discuss how the PIQFC can be used in different operating regimes by changing the pump amplitudes, before giving a detailed explanation of the experimental setup. Next, the authors introduce the cold Rubidium cloud and describe how they can generate a path-entangled pair of an anti-Stokes (AS) photon and an atomic excitation. After transforming the path information into polarization, the AS photon is converted in the PIQFC towards the telecom band. Finally, the atomic excitation is read-out into a Stokes (S) photon and the entanglement is proven by quantum state tomography between the AS and S photons.

The presented work is scientifically interesting and in line with recent progress of polarization independent quantum frequency conversion (see e.g. Refs. 7, 52). The presented way of achieving PIQFC using a Sagnac interferometer is an elegant solution, which however was also very recently employed in another experiment not mentioned by the authors (see arXiv:1710.04866 (2017)). In that experiment polarization entanglement between a trapped Calcium ion and a photon converted to the telecom O-band was demonstrated with high fidelity. Although that matter system is different to the Rubidium ensemble chosen in the current work, both experiments demonstrate polarization entanglement between a matter quantum system and a single photon converted to the telecom band. I highly recommend the authors to take note of the other manuscript and to compare it to their work. In particular, I would like to ask the authors to be careful to not mislead the reader by suggesting that their work represents the only or the first polarization-entanglement between a matter quantum system and a telecom photon interfaced by quantum frequency conversion. Still I agree that the presented work (together with the other experiment) is at the forefront of research in this particular field and represents an important step towards long distance quantum networking using telecom photons.

Apart from my general comment above, I would like to give also some other more technical remarks:

1. I think it is important to make clear that the various configurations of the PIQFC presented in Fig.1 and discussed in the text ((a) half-BS, (b) PBS, (c) PPBS) are rather potential modes of operation, than actual implementations in the current experiment.

2. I would appreciate, if the authors could give some more technical information (e.g. in Supplementary Information), like specifications of the QFC chip (PE/Ridge waveguide, length, phase-matching temperature, ...) or the transmission losses of several optical elements (e.g. dichroic mirrors, waveguide, filtering cavities, etalon, FBG, ...) etc..
3. How is the QFC efficiency exactly defined? (measured from where to where? in fiber or not?)
4. If I understand correctly, the filtering (i.e. coated lens, etalon, FBG) is just used if the PIQFC is interfaced with the MOT. It would be useful to know which noise characteristics (Signal to Noise/Background Ratio) the PIQFC exhibits in bare and filtered configurations, to be able to compare it to other approaches.
5. Some further characterization measurements of the PIQFC would be interesting as well. For example, which visibilities or level of polarization-crosstalk is achievable depending on the input polarization?
6. The authors should explain more in detail, the preparation and coupling scheme used to operate the cold Rubidium cloud. For example, is there a magnetic field to define a quantization axis? In which Zeeman sub-states are the atoms prepared or is the population equally distributed?
7. I found it a bit surprising that the authors do neither cite nor even mention the well-known name of the protocol used to generate the entanglement (DLCZ, Nature 414, 413 (2001)).
8. What is the currently achievable storage time of the spin excitation stored in the atomic cloud? To which extend cloud the angle between the +/- paths and the classical write/read beams reduced to extend the storage time without significantly diminishing the entanglement?
9. It would be good scientific practice to give the calculated values of the reconstructed density matrices in the Supplementary Information.

In conclusion, I believe that the presented manuscript can be an adequate fit for Nature Communications once the above issues are convincingly addressed and revised in the manuscript.

Reviewer #2 (Remarks to the Author):

Ikuta and co-workers propose a manuscript reporting on entanglement between single atomic excitations and telecom photons. Atom-photon entanglement is first created using off-resonant spontaneous Raman processes. The photon wavelength is subsequently converted to the telecom band using a non-linear waveguide. Entanglement between atoms and telecom photons is finally detected by mapping the atomic excitation into photons and realising the tomography of the resulting two-photon state. The experiment is fairly new. The manuscript is well written and the

results are well presented. I would suggest publication in Nature Communications provided that the authors answer the following questions/comments.

1-As mentioned by the authors, frequency conversion of photons emitted from cold Rb atoms has been reported in 2 publications recently. They claim that the quantum state preservation has not been shown in these 2 publications. I think that this claim is not fair. Ref. 28 reported on the non-classicality of atom-telecom photon correlations through the Cauchy-Schwarz criterion. This does not imply entanglement but is a form of non-classical state. I thus suggest that the authors either provide more comments or simply change their claim.

2-The authors claim that long-lived quantum memories and atom-telecom photon entanglement are central tools for quantum networks. I do agree with this statement and would recommend to cite relevant publications to justify this claim. However, the way entanglement is generated in the reported experiment is probabilistic, that is atom-photon entanglement is created probabilistically. At each experiment run, there is every chance that no anti-Stokes photon is emitted. There is also a chance that more than 1 anti-Stokes is produced. Can the author comments on the usefulness of their concrete approach for quantum networks? Is there a known protocol that can be used with a probabilistic single atomic excitation -- telecom polarisation photon entanglement? Along the same line, what is the probability of creating atom-photon entanglement for the reported fidelities? is this the main limitation for the purity?

3-From a theoretical point of view, a frequency conversion is a unitary process, thus performing with unit efficiency. This is not what is shown in Fig. 1 e and an explanation on why the efficiency saturates with the pump power would be useful.

4-How would the atom-telecom photon correlations vary as a function of the storage times? What is the longest storage time achievable to keep the detection of photon-photon entanglement?

Reviewer #3 (Remarks to the Author):

The manuscript presents experimental results in which a Rb atomic ensemble is entangled with the polarization degree of freedom of an infrared photon, and that photon is subsequently converted to

a telecom wavelength (1522 nm) in a polarization insensitive manner. Entanglement between the Rb ensemble and the telecom photon is demonstrated. Similar entanglement with polarization-preserving frequency conversion to telecom has previously been demonstrated via four-wave mixing (Ref. 3), but this is the first demonstration using solid-state quantum frequency conversion (along with the work reported in a similar manuscript, also submitted to Nature Communications). As such, it represents a significant step towards quantum networks that are capable of linking together quantum memories over long distances. I believe that the manuscript will be of interest not only to researchers working on quantum memories and frequency conversion but to the larger quantum information science community. This approach of using a Sagnac interferometer (also implemented by the other research team) may become a standard approach in other research groups.

The results are clearly and logically presented and convincing and in my opinion merit publication in Nature Communications. The most significant shortfall I see is that the authors do not discuss what limits the fidelity of their entangled states with respect to a maximal entangled state, and (more importantly) they do not address why this fidelity becomes worse when frequency conversion is carried out, and whether it may be possible to remedy this in the future. Such a discussion would certainly improve the manuscript as it would help the reader understand the limits and potential drawbacks of this polarization-insensitive approach to frequency conversion.

I also would like to raise the following minor points:

In Fig. 1d, elements of the setup (DM2, DM1) are labelled with abbreviations that are only defined later on in the main text and not in the figure caption.

Error bars should be given in Fig. 1e. A single fit is used for both the clockwise and counter-clockwise data, but would it not make more sense to fit the two data sets separately and to discuss their slightly different behavior?

At the top of p. 3 and at the beginning of the discussion on p. 5, "polarizing upper frequency photon at 780 nm" and "polarizing telecom photon" should be "polarized upper frequency..." and "polarized telecom photon."

The authors have the tendency to use the tilde symbol to mean "approximately" even when it should be possible to state exact numbers and to provide the reader with estimates of the uncertainty. For example, on p. 4, there is a quantum efficiency of $\sim 60\%$, bandwidths of ~ 100 MHz, a conversion efficiency of $\sim 30\%$, and a pump power of ~ 0.2 W. Surely it should be possible to provide more precision for these values, especially for the pump power and conversion efficiency.

We thank the reviewer for the valuable comments. We also appreciate that the three reviewers basically recommended our manuscript to be published on Nature Communications. We believe that the revised manuscript will meet the requirement for the publication on Nature Communications.

Reply to the comments of reviewer 1

Q0: The presented way of achieving PIQFC using a Sagnac interferometer is an elegant solution, which however was also very recently employed in another experiment not mentioned by the authors (see arXiv:1710.04866 (2017)). In that experiment polarization entanglement between a trapped Calcium ion and a photon converted to the telecom O-band was demonstrated with high fidelity. Although that matter system is different to the Rubidium ensemble chosen in the current work, both experiments demonstrate polarization entanglement between a matter quantum system and a single photon converted to the telecom band. I highly recommend the authors to take note of the other manuscript and to compare it to their work. In particular, I would like to ask the authors to be careful to not mislead the reader by suggesting that their work represents the only or the first polarization-entanglement between a matter quantum system and a telecom photon interfaced by quantum frequency conversion.

A0: As the reviewer pointed out, the recent paper arXiv:1710.04866 (2017) appeared on the arxiv while we were finalizing our manuscript. In order to address the work and the difference from our work, we add sentences in the last of the main text of the revised manuscript as

“Note added: During preparation of our paper, we found a related work [60] in which polarization entanglement between a trapped calcium ion and a telecom O-band photon with QFC was demonstrated. Similarly to our experiment, they employed a PPLN waveguide in a Sagnac-type interferometer to have the ability to an arbitrary polarization state. A major difference is that our scheme forms a Sagnac interferometer for not only target photons but also the pump light, resulting in the robustness against the phase instability without any feedback control. ”

[60] Bock, M. et al. High-fidelity entanglement between a trapped ion and a telecom photon via quantum frequency conversion. arXiv preprint arXiv:1710.04866 (2017).

Q1: I think it is important to make clear that the various configurations of the PIQFC presented in Fig.1 and discussed in the text ((a) half-BS, (b) PBS, (c) PPBS) are rather potential modes of operation, than actual implementations in the current experiment.

A1: In order to clarify the various configurations of the PIQFC in the main text, we revised the labels (a) half-BS, (b) PBS, (c) PPBS overlapping with the labels in Fig. 1 in the previous manuscript to (I), (II) and (III).

In addition, we added an explanation about the meaning of the transmittance and reflectance of Eqs. (1) and (2) on the fifth line after Eq. (2) as

“... for splitting into the two frequency modes propagating in the same direction ...”

and in 3rd line of the right hand side of page 2, we revised an explanation about the actual implementations in the current experiment and other configurations as

“By compensating the constant phase, we thus achieve polarization insensitive QFC, which we used in the experiments reported in this paper. On the other hand, we may also achieve other modes of operation depending on the choice of R_H and R_V .”

Q2: I would appreciate, if the authors could give some more technical information (e.g. in Supplementary Information), like specifications of the QFC chip (PE/Ridge waveguide, length, phase-matching temperature, ...) or the transmission losses of several optical elements (e.g. dichroic mirrors, waveguide, filtering cavities, etalon, FBG, ...) etc..

A2: We added the detailed information about the PPLN crystal in Methods section. In addition, we also added the transmittance of the optics in Methods section.

Q3: How is the QFC efficiency exactly defined? (measured from where to where? in fiber or not?)

A3: The efficiency was measured from the front to the end of the QFC which were written in the caption of Fig. 1e and the main text in the previous manuscript. In order to give further information about the experiment for estimating the conversion efficiency, we added the detailed information of the PPLN waveguide and how to measure the efficiency in Methods section related to Q2.

- Q4:** If I understand correctly, the filtering (i.e. coated lens, etalon, FBG) is just used if the PIQFC is interfaced with the MOT. It would be useful to know which noise characteristics (Signal to Noise/Background Ratio) the PIQFC exhibits in bare and filtered configurations, to be able to compare it to other approaches.
- A4:** We estimated the amount of the noises at QFC by measuring the photon counts without the input of the signal photons. The count rates were 3.1×10^{-6} counts/pulse and 3.5×10^{-6} counts/pulse for H and V polarization. By considering the quantum efficiency 60 % of the SSPD and transmittances 50 % and 17 % of the frequency filters (the etalon and the FBGs), the noise photons just after the QFC are estimated to be 6.1×10^{-5} counts/pulse and 7.0×10^{-5} counts/pulse for H and V polarization, respectively, in the bandwidth of 540 MHz which are calculated by using the bandwidths 690 MHz and 0.9 GHz of the frequency filters. Since the total conversion efficiency used in our experiment is 0.33 for both H and V polarization, the signal to noise ratios for a deterministic single photon input are 5400 and 4700.
- Q5:** Some further characterization measurements of the PIQFC would be interesting as well. For example, which visibilities or level of polarization-crosstalk is achievable depending on the input polarization?
- A5:** The polarization-crosstalk of the PIQFC is very small and we can hardly see any amount of the polarization-crosstalk. This is because even if the H(V) polarized light is reflected(transmitted) at the PBS, whose extinction ratio is $> 100:1$, in the Sagnac interferometer, the light is coupled to the PPLN crystal with H polarization which does not interact with the type-0 PPLN. In addition, the remaining light coming from the PPLN emerges from the opposite output port of the PBS.
- Q6:** The authors should explain more in detail, the preparation and coupling scheme used to operate the cold Rubidium cloud. For example, is there a magnetic field to define a quantization axis? In which Zeeman sub-states are the atoms prepared or is the population equally distributed?
- A6:** The magnetic field is not applied in the QFC experiment. In this case, the energy level of the Zeeman sub-levels are degenerated, and we assume the atomic states initialized into the ground level g_a are fully mixed over the sublevels. In order to clarify this statement, we added an explanation in caption of Fig. 2b as
 “The magnetic sublevel is degenerated in our experiment due to the absence of the magnetic field, and the polarization selection of write, read, AS and S light is to extract larger amounts of the emitted photons under an assumption that the initialized atomic states in g_a are fully mixed over the magnetic sublevels (Supplemental material in Ref. [29]). ”
- Q7:** I found it a bit surprising that the authors do neither cite nor even mention the well-known name of the protocol used to generate the entanglement (DLCZ, Nature 414, 413 (2001)).
- A7:** On line 3 in the second paragraph in introduction, we revised the manuscript and added the references about quantum repeater protocols and its review as
 “For a long-distance quantum communication with quantum repeater algorithms [30–32],...”
 [30] Briegel, H.-J., Dür, W., Cirac, J. I. & Zoller, P. Quantum repeaters: The role of imperfect local operations in quantum communication. Phys. Rev. Lett. 81, 59325935 (1998).
 [31] Duan, L.-M., Lukin, M., Cirac, J. I. & Zoller, P. Long- distance quantum communication with atomic ensembles and linear optics. Nature 414, 413418 (2001).
 [32] Sangouard, N., Simon, C., de Riedmatten, H. & Gisin, N. Quantum repeaters based on atomic ensembles and linear optics. Reviews of Modern Physics 83, 33 (2011).
- Q8:** What is the currently achievable storage time of the spin excitation stored in the atomic cloud? To which extend cloud the angle between the +/- paths and the classical write/read beams reduced to extend the storage time without significantly diminishing the entanglement?
- A8:** In the current experimental setup of MOT, the storage time is < 500 ns which was estimated by measuring the dependency of the cross correlation function between AS and S photons on the storage time. The angle dependency of the storage time or the fidelity has not been investigated in our experiment. This might be a future experimental issue.
- Q9:** It would be good scientific practice to give the calculated values of the reconstructed density matrices in the Supplementary Information.

A9: We added the calculated values of the reconstructed density matrices $U_{\theta_0}^\dagger \rho_{S,AS} U_{\theta_0}$ and $U_{\theta_1}^\dagger \rho_{S,AS} U_{\theta_1}$ in the Supplementary Information.

Reply to the comments of reviewer 2

Q1: As mentioned by the authors, frequency conversion of photons emitted from cold Rb atoms has been reported in 2 publications recently. They claim that the quantum state preservation has not been shown in these 2 publications. I think that this claim is not fair. Ref. 28 reported on the non-classicality of atom-telecom photon correlations through the Cauchy-Schwarz criterion. This does not imply entanglement but is a form of non-classical state. I thus suggest that the authors either provide more comments or simply change their claim.

A1: In order to review the previous two references (28 and 29) about QFC of photons emitted from Rb atoms clearly, we revised our manuscript on line 11 of the second paragraph in the introduction as

“in which a non-classical correlation between atoms and a telecom photon was observed [28] and a single spin excitation of atoms heralded by a telecom photon detection was observed [29].”

In addition, in order to emphasize the meaning of the quantum memory, we added a phrase in the next sentence as

“But the quantum state preservation, which is an ability never substituted by a classical memory, has never been shown yet.”

Q2: The authors claim that long-lived quantum memories and atom-telecom photon entanglement are central tools for quantum networks. I do agree with this statement and would recommend to cite relevant publications to justify this claim. However, the way entanglement is generated in the reported experiment is probabilistic, that is atom-photon entanglement is created probabilistically. At each experiment run, there is every chance that no anti-Stokes photon is emitted. There is also a chance that more than 1 anti-Stokes is produced. Can the author comment on the usefulness of their concrete approach for quantum networks? Is there a known protocol that can be used with a probabilistic single atomic excitation – telecom polarisation photon entanglement? Along the same line, what is the probability of creating atom-photon entanglement for the reported fidelities? is this the main limitation for the purity?

A2: As the reviewer pointed out, there are non zero probabilities that no photon is emitted and more than one photons are emitted, because the cloud-based quantum memory uses the parametric interaction between boson modes. Typically in order to suppress the multiple photon emission for a high quality of the quantum state, a single photon emission probability p is set much smaller than 1, which means the no photon emission probability is large. But this can be solved by using multiplexing techniques discussed in Refs. [1] and [2]. In the calculation of the supplementary information in Ref.[2], the applicability of their cloud-based quantum memory to the DLCZ protocol with N multiplexing such that $Np = 1$ was discussed. By starting with fidelity 95 % of the initial photon pairs due to the multiple photon emission errors, they showed the possibility to achieve 1000-km entanglement distribution with a rate of sub Hz. Unfortunately, a perfect QFC is assumed in the simulation. We guess noise effects induced by QFC can be cancelled by using a smaller value of p , and the smaller probability and non unit conversion efficiency can be covered by a larger value of N for multiplexing.

[1] Sangouard, N., Simon, C., de Riedmatten, H. & Gisin, N. Quantum repeaters based on atomic ensembles and linear optics. *Reviews of Modern Physics* 83, 33 (2011).

[2] Yang, S.-J., Wang, X.-J., Bao, X.-H. & Pan, J.-W. An efficient quantum light-matter interface with sub-second lifetime. *Nature Photonics* 10, 381384 (2016).

Q3: From a theoretical point of view, a frequency conversion is a unitary process, thus performing with unit efficiency. This is not what is shown in Fig. 1 e and an explanation on why the efficiency saturates with the pump power would be useful.

A3: One of the reasons for the non-unit efficiency is that the coupling efficiencies of the signal light are not 1, which are estimated to be 0.83 and 0.65 for CW and ACW directions, respectively. Another reason may be mode mismatching of the signal and the pump light propagating in the PPLN waveguide. In order to explain these experimental imperfections, we added the experimental details of the conversion efficiencies in the Methods section “The PPLN waveguide in this experiment.”

- Q4:** How would the atom-telecom photon correlations vary as a function of the storage times? What is the longest storage time achievable to keep the detection of photon-photon entanglement?
- A4:** We measured the dependency of the cross correlation function between the S and the AS photons on the storage time. The cross correlation became almost 1 for the storage time of 500 ns. The entanglement measurement takes a lot of time in our current experimental setup, and it is difficult to perform the quantum state tomography for various storage times. Instead, we roughly calculated the expected storage time for obtaining the fidelity > 0.5 after QFC, which guarantees the existence of the entanglement, to be ~ 250 ns with the following assumptions and the experimental results of the cross correlation functions:
- (1) The cross correlation function follows the exponential decay and converges to 1.
 - (2) The fidelity to the maximally entangled state is described by $(1 + |V_z| + |V_x| + |V_y|)/4$ by using the visibilities V_i of the two-photon polarization state which are measured in the H/V polarization basis for V_z , diagonal/anti-diagonal polarization basis for V_x , and left/right circular polarization basis for V_y .
 - (3) $V_x = V_y = 3V_z/4$ is always satisfied for any storage time which is estimated from the current experimental result.
 - (4) $V_z = (g_{S,ASt|HH}^{(2)} + g_{S,ASt|VV}^{(2)} - 2)/(g_{S,ASt|HH}^{(2)} + g_{S,ASt|VV}^{(2)} + 2)$ is satisfied by assuming that $g_{S,ASt|HV}^{(2)} = g_{S,ASt|VH}^{(2)} = 1$ and the single counts do not depend on the measurement polarization basis.

Reply to the comments of reviewer 3

- Q1:** The most significant shortfall I see is that the authors do not discuss what limits the fidelity of their entangled states with respect to a maximal entangled state, and (more importantly) they do not address why this fidelity becomes worse when frequency conversion is carried out, and whether it may be possible to remedy this in the future. Such a discussion would certainly improve the manuscript as it would help the reader understand the limits and potential drawbacks of this polarization-insensitive approach to frequency conversion.
- A1:** The main cause of the degradation of the fidelity *without* QFC is the non-negligible contamination from the 850-nm light used for the stabilization of the interferometer by a mirror on a Piezo stage (PZT) as mentioned in the caption of Fig. 2a. On the other hand, the fidelity *with* QFC does not suffer: the 850-nm light is eliminated because it does not satisfy the phase matching of the PPLN crystal or the converted light cannot pass through the frequency filters even if the light is converted. Instead, it suffers from the Raman scattering noise photons induced at QFC due to the strong pump light. In order to estimate the effects of the noise photons and discuss how to improve the fidelity after QFC, we added a discussion paragraph before the conclusion paragraph.
- Q2:** Error bars should be given in Fig. 1e. A single fit is used for both the clockwise and counter-clockwise data, but would it not make more sense to fit the two data sets separately and to discuss their slightly different behavior?
- A2:** As we added an explanation about the experimental details of the conversion efficiency in Methods section, the measurement was performed by using a classical laser light at 780 nm and a power meter. The powers of the input light and pump light were about 5 mW and < 300 mW. In these cases the typical error of the power meter was about 1% which we confirmed in the experiment. Similarly to typical photonics experiments, such errors are smaller than the markers of data.
- About the fitting curve for the experimental data, as the reviewer pointed out, the behavior of the two data set for CW and ACW directions are slightly different. We guess the reasons are the difference of the coupling efficiencies of the signal light to the PPLN and that of the mode mismatching of the signal and the pump light propagating in the PPLN waveguide. We added the details in Methods section “The PPLN waveguide in this experiment.” In addition, in Fig. 1e, we added two curves fitted to them separately, and added an explanation of the two curves in the caption of Fig. 1e.
- Q3:** At the top of p. 3 and at the beginning of the discussion on p. 5, “polarizing upper frequency photon at 780 nm” and “polarizing telecom photon” should be “polarized upper frequency...” and “polarized telecom photon.”
- A3:** We revised our manuscript as the reviewer pointed out.
- Q4:** The authors have the tendency to use the tilde symbol to mean “approximately” even when it should be possible to state exact numbers and to provide the reader with estimates of the uncertainty. For example, on p. 4, there is a quantum efficiency of $\sim 60\%$, bandwidths of ~ 100 MHz, a conversion efficiency of $\sim 30\%$, and a pump

power of ~ 0.2 W. Surely it should be possible to provide more precision for these values, especially for the pump power and conversion efficiency.

A4: We modified the approximation values about the FWHMs of the monolithic cavity-coated lenses and the etalon, the conversion efficiencies and the pump powers for CW and ACW directions to the observed values as precisely as we could. About the fiber Bragg gratings and the quantum efficiencies of the photon detectors, we could not measure the FWHM and the quantum efficiencies precisely, and we left the approximation values from the spec sheets and the reported paper.

Reviewers' comments:

Reviewer #1 (Remarks to the Author):

First of all, I would like to thank the authors for their carefully updated re-submission.

I found the rebuttal letter quite complete, addressing my questions in a comprehensive and convincing way. In almost all open points, the authors updated their manuscript accordingly. Just my question about the noise characteristics (Q4 & A4) was not included in the updated version of the manuscript. So my only recommendation here is, that the authors should somehow make this information (noise count rates, SNR etc..) accessible to the readers as well, as these values are usually key figures of merit in such experiments.

Apart from this minor comment I'm fine with the manuscript in its current form and support publication in Nature Communications.

Reviewer #2 (Remarks to the Author):

The authors have answered my concerns correctly. I thus propose to publish their work in Nature Communications.

Reviewer #3 (Remarks to the Author):

Concerning the first point addressed by the authors, I find it helpful that a discussion has now been included (in the second-to-last-paragraph) on the photon correlations with and without QFC conversion. However, the relationship between the correlation rates and entangled state fidelities isn't clearly explained. Moreover, the authors claim at the end of this section that "the fidelity after QFC could be near the value with QFC." I find this confusing, as the authors have explained that the fidelity after QFC doesn't suffer from contamination with 850 nm light. Shouldn't it then be possible for the measured fidelity after QFC to be higher than the measured fidelity before QFC? In short, I don't feel that the question "What limits the fidelity of the entangled states (after frequency conversion)?" has been satisfactorily answered.

Concerning the second point, I understand that the error bars are not plotted because they are smaller than the markers, but I nevertheless think that this should be indicated in the figure caption.

Otherwise, all of the points raised in my original review have been addressed.

We thank the reviewers for the further valuable comments. We also appreciate that the three reviewers recommended our manuscript to be published on Nature Communications. We believe that the revised manuscript will meet the requirement for the publication on Nature Communications.

Reply to the comments of reviewer 1

We added the discussion about the noise characteristics of the QFC in the Methods section.

Reply to the comments of reviewer 3

Q1: However, the relationship between the correlation rates and entangled state fidelities isn't clearly explained.

A1: We revised the manuscript to clarify the relation. The measurements of cross correlation functions were made in order to determine the amount of noise photons present in the actual experiment. On the other hand, the three types of visibilities $V_z(\rho) = \text{tr}(\rho ZZ)$, $V_x(\rho) = \text{tr}(\rho XX)$ and $V_y(\rho) = \text{tr}(\rho YY)$ were introduced, which are straightforwardly connected to the fidelity as $F(\rho) = (1 + V_z(\rho) + V_x(\rho) - V_y(\rho))/4$. Under a few assumptions for simplicity, we can derive how the visibilities change depending on the amount of noise photons (which is described in Methods). This allows us to predict how the fidelity will change as the amount of noise photons varies.

Q2: Moreover, the authors claim at the end of this section that "the fidelity after QFC could be near the value with QFC." I find this confusing, as the authors have explained that the fidelity after QFC doesn't suffer from contamination with 850 nm light. Shouldn't it then be possible for the measured fidelity after QFC to be higher than the measured fidelity before QFC? In short, I don't feel that the question "What limits the fidelity of the entangled states (after frequency conversion)?" has been satisfactorily answered.

A2: Similarly to the estimation of the amount of the noise photons induced at QFC which is written in the 1st revision manuscript, we found the estimation of the amount of the 850-nm light without QFC is possible by comparing the experimental results with and without 850-nm light. From the estimation and using the cross correlation functions, we calculated visibilities and a fidelity F^* without QFC which would be obtained if we could remove the 850-nm laser without affecting the phase stability. The comparison of the visibilities and those with QFC obtained by the main experimental results implies that the fidelity after QFC is limited due to the noise photons induced at QFC and decoherence between H- and V-polarized photons. In the best case, when all of the adverse effects with QFC could be removed, the fidelity after QFC could be near the fidelity F^* .

In Discussion section of the revised manuscript, we added the above discussion.

Q3: Concerning the second point, I understand that the error bars are not plotted because they are smaller than the markers, but I nevertheless think that this should be indicated in the figure caption.

A3: We added the explanation in the caption of Fig.1e as

"The error of the power meter was about 1% and they are smaller than the markers of data."